# Recording γ-secretase activity in living mouse brains

**Steven S Hou*†, Yuya Ikegawa†, Yeseo Kwon, Natalia Wieckiewicz, Mei CQ Houser, Brianna Lundin, Brian J Bacskai, Oksana Berezovska, Masato Maesako***

MassGeneral Institute for Neurodegenerative Disease, Massachusetts General Hospital, Harvard Medical School, Charlestown, United States

**Abstract** γ-Secretase plays a pivotal role in the central nervous system. Our recent development of genetically encoded Förster resonance energy transfer (FRET)-based biosensors has enabled the spatiotemporal recording of γ-secretase activity on a cell-by-cell basis in live neurons *in culture*. Nevertheless, how γ-secretase activity is regulated *in vivo* remains unclear. Here, we employ the near-infrared (NIR) C99 720–670 biosensor and NIR confocal microscopy to quantitatively record γ-secretase activity in individual neurons in living mouse brains. Intriguingly, we uncovered that γ-secretase activity may influence the activity of γ-secretase in neighboring neurons, suggesting a potential 'cell non-autonomous' regulation of γ-secretase in mouse brains. Given that γ-secretase plays critical roles in important biological events and various diseases, our new assay *in vivo* would become a new platform that enables dissecting the essential roles of γ-secretase in normal health and diseases.

**\*For correspondence:**
shou@mgh.harvard.edu (SSH);
MMAESAKO@mgh.harvard.edu
(MM)

†These authors contributed
equally to this work

**Competing interest:** The authors
declare that no competing
interests exist.

**Reviewing Editor:** John R
Huguenard, Stanford University
School of Medicine, United
States

## eLife assessment

Hou and colleagues describe the the use of a previously characterized FRET sensor for use in determining γ-secretase activity in the brain of living mice. In an approach that targeted the sensor to neurons, they observe patterns of fluorescent sensor readout suggesting clustered regions of secretase activity. These results once validated would be **valuable** in the field of Alzheimer's Disease research, yet further validation of the approach is required, as the current evidence provided is **inadequate** to support the conclusions.

## Introduction

γ-Secretase is an intramembrane aspartyl protease complex responsible for the proteolytic processing of a wide range of transmembrane proteins (**Güner and Lichtenthaler, 2020**) such as Notch receptors (**De Strooper et al., 1998**) and APP (**De Strooper et al., 1999**). γ-Secretase is ubiquitously expressed in various tissues and plays multifunctional roles in essential biology and diseases. For instance, the mice lacking presenilin (PSEN) 1 and 2, the catalytic component of γ-secretase (**Wolfe et al., 1999**), are nonviable due to abnormal neurogenesis and vasculature and skeletal formation (**Shen et al., 1997**; **Wong et al., 1997**). Notch knockout (KO) mice recapitulate lethal phenotypes of the PSEN1/2 KO (**Swiatek et al., 1994**; **Huppert et al., 2000**), suggesting the essential role of γ-secretase-mediated Notch processing in normal development. Aberrant Notch processing also causes several types of cancer (**Allenspach et al., 2002**; **Aster et al., 2017**; **McCaw et al., 2021**) as well as a chronic skin inflammatory disease - Hidradenitis suppurativa (HS) (**Wang et al., 2010**; **Wang et al., 2021**).

γ-Secretase plays an essential role in the central nervous system. Conditional knockout of PSEN1 and 2 (**Saura et al., 2004**) or other components of γ-secretase in excitatory (**Tabuchi et al., 2009**;

*Acx et al., 2017*; *Bi et al., 2021*), as well as inhibitory neurons (*Kang and Shen, 2020*), leads to age-dependent neuronal loss in adult mice. Mutations in PSEN genes lead to early-onset familial Alzheimer's disease (AD) (*Sherrington et al., 1995*; *Levy-Lahad et al., 1995*) and frontotemporal dementia (FTD) (*Raux et al., 2000*; *Dermaut et al., 2004*). Some PSEN mutations are linked to epilepsy (*Mann et al., 2001*; *Velez-Pardo et al., 2004*). γ-Secretase is also responsible for the generation of β-amyloid (Aβ) peptides (*De Strooper et al., 1999*), the accumulation of which in the brain parenchyma is one of the pathological hallmarks of AD. Although exactly how is still a matter of debate, it is highly plausible that changes in γ-secretase activity could be ultimately linked to the development and progression of neurodegenerative diseases such as AD.

Our recent development of the genetically encoded C99 YPet-mTurquoiseGL (C99 Y-T) biosensor has, for the first time, allowed recording γ-secretase activity over time, on a cell-by-cell basis, in live neurons *in culture*. Using the C99 Y-T biosensor, we uncovered that γ-secretase activity is heterogeneously regulated among mouse cortex primary neurons (*Maesako et al., 2020*). Nevertheless, to investigate the role of γ-secretase in-depth, an assay that permits investigation of the dynamics of γ-secretase in more physiological conditions *in vivo* would be required. In the conventional γ-secretase activity assay, APP or Notch1-based recombinant substrates and isolated γ-secretase-rich membrane fractions from rodent/human brains are incubated *in vitro*, followed by the detection of either Aβ peptides or the intracellular domain fragment (*Hoke et al., 2005*; *Kakuda et al., 2006*; *Szaruga et al., 2015*). Although the assay is highly sensitive, it only reports γ-secretase activity in a bulk population of cells *ex vivo*, and neither provides spatiotemporal nor cell-by-cell information about γ-secretase activity in live cells *in vivo*.

To overcome these shortcomings and fulfill the detection capability of γ-secretase activity *in vivo*, we have recently developed a near-infrared (NIR) analog: C99 720–670 biosensor (*Houser et al., 2020*), since the NIR spectral region exhibits greater depth penetration and minimal absorption, and this spectral region exhibits low autofluorescence. In this study, we employed AAV delivery of the C99 720–670 biosensor and NIR confocal microscopy to monitor γ-secretase activity in intact mouse brains. This study, for the first time, reports successful recording of γ-secretase activity on a cell-by-cell basis in live neurons *in vivo*, as opposed to an existing assay in a bulk population of cells *ex vivo*. Moreover, it highlights the feasibility of NIR confocal microscopy for monitoring biological events in live mouse brains. In agreement with the previous study *in vitro* (*Maesako et al., 2020*), we uncovered that γ-secretase activity is differently regulated in individual neurons in live mouse brains *in vivo*. Although the effect size is modest, we also found a statistically significant correlation between γ-secretase activity within a neuron and the activity of γ-secretase in neighboring neurons, suggesting a potential 'cell non-autonomous' regulation of γ-secretase activity in mouse brains. The new *in vivo* imaging platform would help to better understand the spatiotemporal regulation of γ-secretase and its consequences in normal health and disease-relevant conditions.

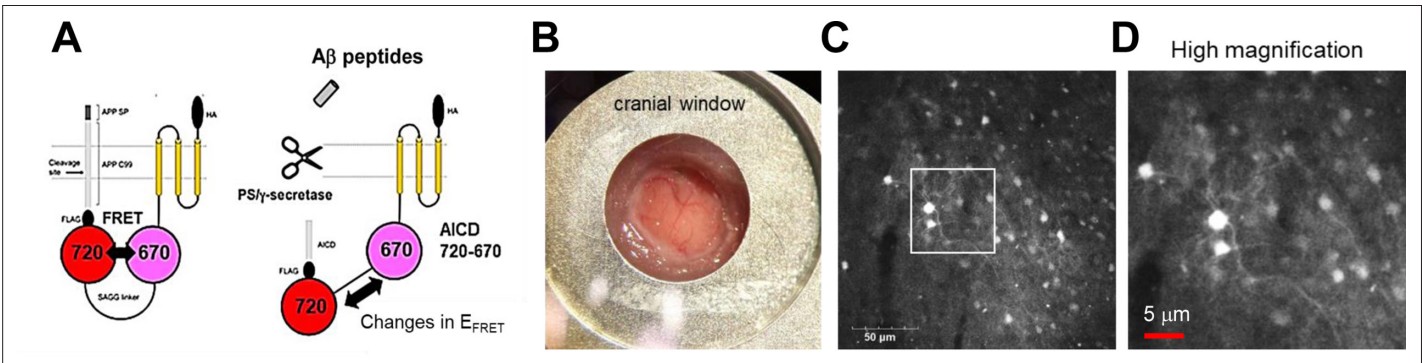

**Figure 1.** Expression of the C99 720–670 biosensor in living mouse brains. (**A**) A schematic presentation of the C99 720–670 biosensor. (**B**) A cranial window was implanted on the top of the brain for image acquisition. (**C**) Extensive expression of the C99 720–670 biosensor in the somatosensory cortex was verified by confocal microscopy *in vivo*. Scale bar: 50 μm. (**D**) A high-magnification image corresponding to the square in **C**. Scale bar: 5 μm.

The online version of this article includes the following figure supplement(s) for figure 1:

**Figure supplement 1.** Z-section images of the brain expressing the C99 720–670 biosensor.

**Figure supplement 2.** Immunohistochemistry of the brain expressing the C99 720–670 biosensor.

## Results

### Expression of the C99 720-670 biosensor in the somatosensory cortex of living mice

We have recently developed genetically encoded FRET-based biosensors that, for the first time, allow quantitative recording of γ-secretase activity on a cell-by-cell basis in cultured neurons (*Maesako et al., 2020*; *Houser et al., 2020*). The principle of the biosensors is that the C-terminus of APP C99, an immediate substrate of γ-secretase, is fused to FRET donor and acceptor fluorescent proteins with a flexible linker, and the sensing domains are stabilized near the membrane by connecting to a membrane-anchoring domain. The proteolytic processing of C99 within the biosensor by endogenous γ-secretase results in a change in the proximity and/or orientation between the donor and acceptor fluorophores, which can be recorded as altered FRET efficiency (*Figure 1A*). Of note, we ensured that the fusion of the donor/acceptor fluorescent proteins and the anchor domain does not significantly affect the cleavage efficiency of C99 by γ-secretase, which was evaluated by comparing the processing efficiency between the cells expressing C99 FLAG and FRET sensor (*Maesako et al., 2020*; *Houser et al., 2020*). To elucidate the cell-by-cell regulation of γ-secretase activity in live mouse brains *in vivo*, we administrated AAV particles packaging the C99 720–670 biosensor under human synapsin promoter into the somatosensory cortex of 4–6 months old C57BL/6 mice implanted with cranial window for imaging (*Figure 1B*). 3–4 weeks post AAV injection, we successfully detected the neurons expressing the C99 720–670 sensor using a confocal microscope with single-photon excitation at 640 nm wavelength and emission at 700–800 nm range (*Figure 1C and D*). Notably, the fluorescence signal from the C99 720–670 biosensor could be detected from the brain surface to approximately 100 µm depth (*Figure 1—figure supplement 1*). Furthermore, immunohistochemical analysis revealed that approximately 40% of NeuN-positive neurons express the C99 720–670 biosensor (*Figure 1—figure supplement 2A and B*), and almost all of the C99 720–670 expressing cells are NeuN-positive but GFAP or Iba-1-negative (*Figure 1—figure supplement 2A and C*). These results suggest that, whereas not all neurons express the C99 720–670 biosensor as expected, the C99 720–670 sensor is expressed in neurons.

### Unbiased image processing of neurons expressing the C99 720-670 biosensor

In image processing (*Figure 2A*), the background fluorescence was first subtracted, and the noise was reduced in the images using median filtering to perform the automatic segmentation of individual neurons. Then, an initial set of ROIs was automatically created on all cellular structures (i.e. cell bodies) in the images using 3D iterative thresholding. Next, the intensities of donor (miRFP670) and acceptor (miRFP720) fluorescence were measured, and the acceptor over donor emission ratios (i.e. 720/670 ratios) were calculated for each ROI. Of note, we uncovered that those ROIs contained non-specific autofluorescent objects whose size and morphology clearly differed from neurons (*Figure 2—figure supplement 1A*). Therefore, morphological filtering was first applied to exclude the non-specific objects from ROIs. Moreover, we analyzed the scatter plot of 720/670 ratios (Y-axis: miRFP670 emission, X-axis: miRFP720 emission) and found that the autofluorescent objects have a significantly lower 720/670 ratio compared to neurons expressing C99 720–670 (*Figure 2—figure supplement 1B*). We also confirmed the identity of the autofluorescent objects by imaging using the same microscope settings in mice without AAV-hSyn1-C99 720–670 injection and found the same population of non-specific objects with low 720/670 ratios is present. Therefore, in the segmentation procedure, we excluded the ROIs exhibiting 720/670 ratios below a set threshold value as 1.5.

To ensure the C99 720–670 biosensor sensitivity *in vivo*, we compared the 720/670 ratios before and after subcutaneous administration of DAPT (a potent γ-secretase inhibitor) (*Dovey et al., 2001*) in the same mouse and cell populations. We found that DAPT dose-dependently increased APP C-terminal fragments (APP CTFs), immediate substrates of γ-secretase. In contrast, full-length APP levels were not significantly altered (*Figure 2—figure supplement 2A*), evidencing the inhibition of endogenous γ-secretase in the brain by DAPT. Whereas we previously found that 720/670 ratios are significantly increased by DAPT in various cell types such as CHO cells and mouse cortical primary neurons (*Houser et al., 2020*; *Maesako et al., 2022*), we found that of DAPT significantly decreases 720/670 ratios in mouse brains *in vivo* (22.1% difference between pre- and post-DAPT administration;

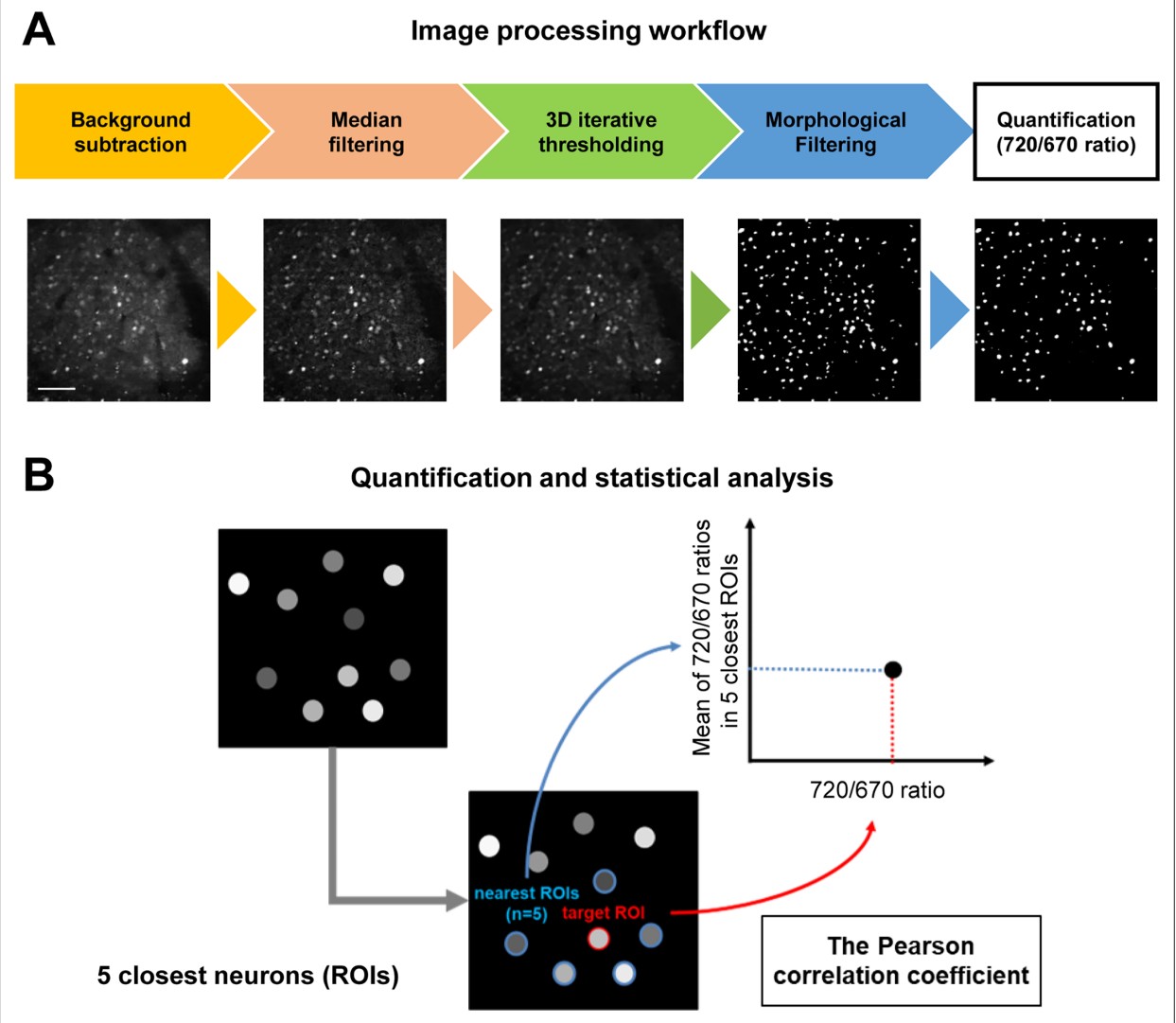

**Figure 2.** Imaging processing workflow and data analysis. (**A**) Before measuring the acceptor over donor emission ratios (i.e., 720/670 ratios) on a cell-by-cell basis, which reports γ-secretase activity in individual neurons, four-step image processing steps were applied: (1) background subtraction, (2) median filtering, (3) 3D iterative thresholding, and (4) morphological filtering. Scale bar: 50 μm. (**B**) To elucidate the relationship between γ-secretase activity and those in neighboring neurons, the distance between neuron and neuron was first determined, then identified each neuron's five closest neurons, and calculated the average 720/670 ratio of the five neighboring neurons. The Pearson correlation coefficient between each neuron's 720/670 ratio and the average ratio of the five neighboring neurons was calculated.

The online version of this article includes the following source data and figure supplement(s) for figure 2:

**Figure supplement 1.** Identification and removal of auto fluorescent objects.

**Figure supplement 1—source data 1.** Numerical source data for *Figure 2—figure supplement 1B*.

**Figure supplement 2.** Validation of the C99 720–670 biosensor in the brain using γ-secretase inhibitor.

**Figure supplement 2—source data 1.** Uncropped and labeled gels for *Figure 2—figure supplement 2A*.

**Figure supplement 2—source data 2.** Raw unedited gels for *Figure 2—figure supplement 2A*.

**Figure supplement 2—source data 3.** Numerical source data for *Figure 2—figure supplement 2C*.

*Figure 2—figure supplement 2B and C*). FRET efficiency generally depends on the proximity and orientation of donor and acceptor fluorescent proteins, and our new finding suggests that orientation plays a significant role in our γ-secretase FRET biosensor. Whether the FRET ratio is increased or decreased by the γ-secretase-mediated biosensor cleavage appears to be dependend on cell types, and therefore must be validated on a model-by-model basis.

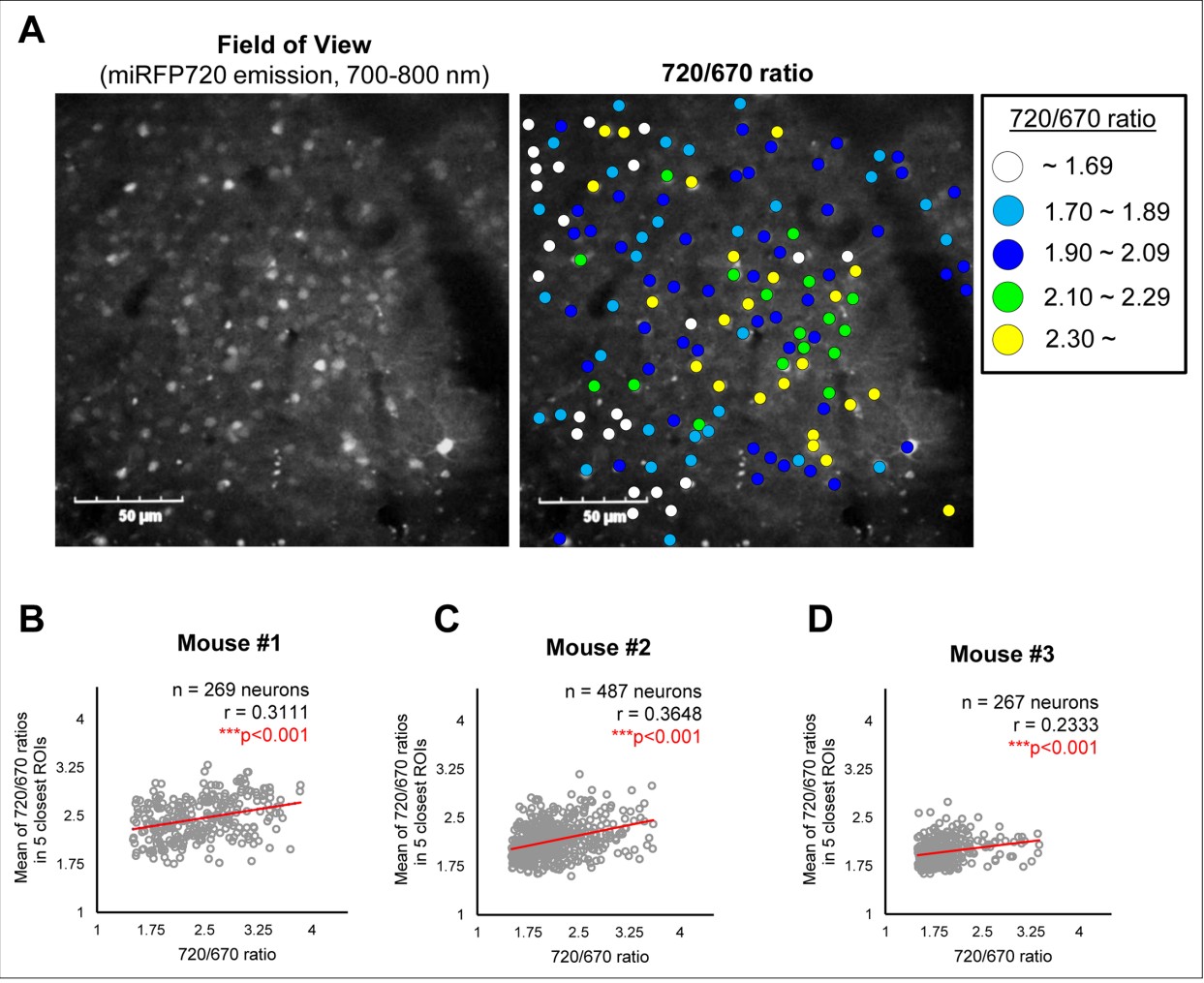

**Figure 3.** A potential 'cell non-autonomous' regulation of γ-secretase in live mouse brains. (**A**) A representative image showing the expressions of the C99 720–670 biosensor (Field of view) and a pseudo-color image corresponding 720/670 ratios (Pseudo-color FRET). Scale bar: 50 μm. (**B–D**) Scatter plots showing the 720/670 ratio in individual neurons (X-axis) and the average Mean of the 720/670 ratio in five neighboring neurons in three independent mice. The number of neurons, correlation coefficient (**r**), and p-value are shown. Pearson correlation coefficient. *** $p < 0.001$.

The online version of this article includes the following source data and figure supplement(s) for figure 3:

**Source data 1.** Numerical source data for *Figure 3B–D*.

**Figure supplement 1.** Expression pattern of the C99 720–670 biosensor.

**Figure supplement 1—source data 1.** Numerical source data for *Figure 3—figure supplement 1A and B*.

**Figure supplement 2.** Validation #1 A potential 'cell non-autonomous' regulation of γ-secretase in mouse brains.

**Figure supplement 2—source data 1.** Numerical source data for *Figure 3—figure supplement 2B and C*.

## 'Cell non-autonomous' regulation of γ-secretase in mouse brains

To elucidate how γ-secretase activity is regulated on a cell-by-cell basis in mouse cortex *in vivo*, we first measured the distance between pairs of neurons, identified each neuron's five closest neurons, and calculated the average 720/670 ratio of the five neighboring neurons. Then, the Pearson correlation coefficient was measured to determine if there is a linear correlation between the 720/670 ratio (as a measure of γ-secretase activity) in each neuron and the average ratio of the five neighboring neurons (*Figure 2B*). Intriguingly, we found that neighboring neurons likely exhibited similar 720/670 ratios (i.e. 'clustering' of the neurons displaying similar 720/670 ratios) (*Figure 3A*). Indeed, while the effect is modest, unbiased quantification and statistical analysis showed a significant linear correlation between the 720/670 ratio in each neuron and the average ratio in five neighboring neurons (*Figure 3B*). Such a statistically significant positive correlation was also detected in two different mice

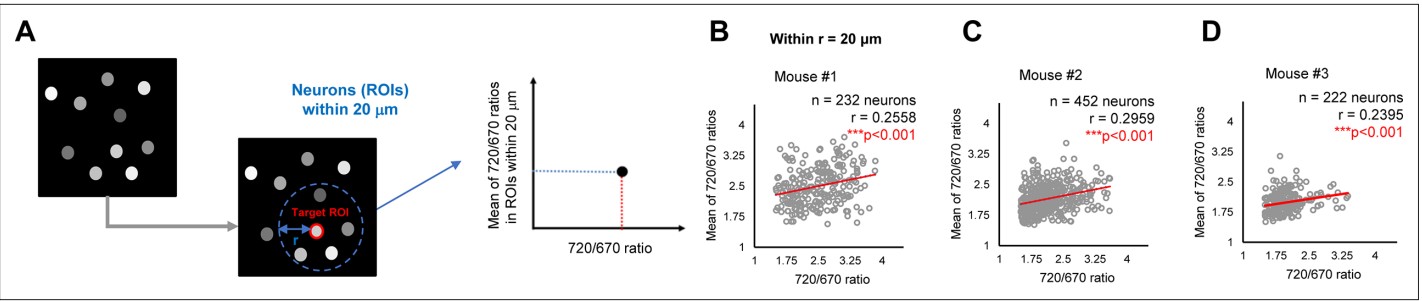

**Figure 4.** Validation #2 A potential "cell non-autonomous" regulation of γ-secretase in mouse brains. (**A**) The average 720/670 ratio of neurons within a 20 µm radius was calculated and plotted. (**B–D**) There was a significant positive correlation between the 720/670 ratio and the average ratio of neurons within a 20 µm radius in three independent mice. The number of neurons, correlation coefficient (r), and p-value are shown. Pearson correlation coefficient. *** p<0.001.

The online version of this article includes the following source data for figure 4:

**Source data 1.** Numerical source data for *Figure 4B–D*.

(*Figure 3C and D*). These results suggest that γ-secretase activity in a neuron may positively correlate with the degree of activity in its neighboring neurons. We found that the C99 720–670 biosensor expression, as measured by miRFP670 emission, positively correlates with those in five neighboring neurons (*Figure 3—figure supplement 1A*), indicating that the AAV was unevenly transduced. However, the 720/670 ratio (i.e. γ-secretase activity) is not correlated with miRFP670 fluorescence intensity (i.e. C99 720–670 biosensor expression) (*Figure 3—figure supplement 1B*), suggesting that, while C99 720–670 biosensor expression was not evenly distributed in the brain, such sensor expression pattern did not impact the capability of γ-secretase recording. We also ensured that the 720/670 ratio was positively correlated with the average 720/670 ratio of the two and ten neighboring neurons (*Figure 3—figure supplement 2A–C*). To further corroborate these findings, we determined the average 720/670 ratio of neurons within a 20 µm radius and examined the correlation between the 720/670 ratio in each neuron and the average 720/670 ratio in neighboring neurons (*Figure 4A*). We found a significant linear correlation between the 720/670 ratio in each neuron and the average ratio of neurons within 20 µm radius in three independent mice (*Figure 4B–D*), further evidencing neurons displaying similar 720/670 ratios co-localize in mouse brains. Lastly, we examined if inhibition of γ-secretase can cancel the clustering of the neurons exhibiting similar levels of γ-secretase activities. As such, we administrated 100 mg/kg DAPT into mice expressing the C99 720–670 biosensor,

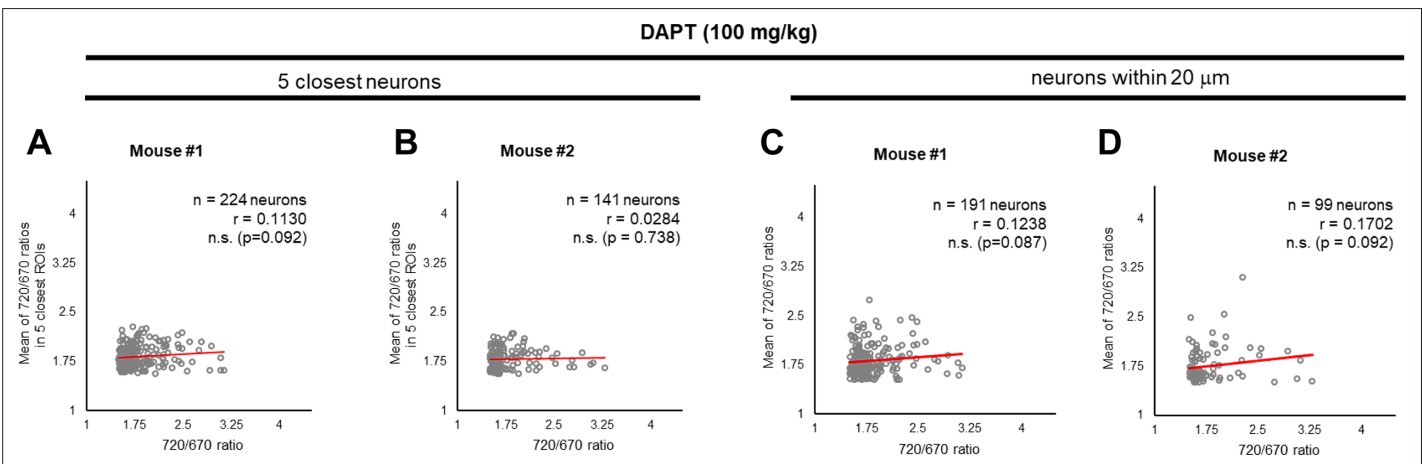

**Figure 5.** γ-Secretase inhibition cancels the 'cell non-autonomous' regulation. (**A, B**) Neither significant correlation between the 720/670 ratio and the average ratio of five closest neurons, nor (**C, D**) the average ratio of neurons within a 20 µm radius was detected after administration of DAPT, a potent γ-secretase inhibitor. The number of neurons, correlation coefficient (r), and p-value are shown. Pearson correlation coefficient. n.s. not significant.

The online version of this article includes the following source data for figure 5:

**Source data 1.** Numerical source data for *Figure 5A–D*.

performed confocal microscopy, adapted the same imaging processing, and performed correlation analysis between the 720/670 ratio in each neuron and the average ratio of the five neighboring neurons or the neurons within 20 μm radius. Notably, there was no significant correlation in DAPT-administrated mice (*Figure 5A–D*), suggesting that the positive correlation between the 720/670 ratio in each neuron and the average ratio in neighboring neurons is canceled by the inhibition of γ-secretase and thus the positive correlation is dependent on γ-secretase activity. Collectively, these results strongly indicate that γ-secretase activities are synchronized in neighboring neurons, and γ-secretase activity may be 'cell non-autonomously' regulated in living mouse brains.

## Discussion

The brain is one of the tissues in which γ-secretase complexes play vital roles. For example, γ-secretase generates Aβ peptides (*De Strooper et al., 1999*), which are accumulated in AD brains. Recent AD clinical trials show that antibodies against Aβ can slow cognitive decline in a statistically significant manner (*van Dyck et al., 2023*; *Sims et al., 2023*). Moreover, γ-secretase is also responsible for maintaining neuronal survival (*Saura et al., 2004*; *Tabuchi et al., 2009*; *Acx et al., 2017*; *Bi et al., 2021*). However, little is known about how endogenous γ-secretase activity is spatiotemporally regulated in the intact brain. In the present study, we recorded γ-secretase activity in individual neurons in a living mouse brain by employing AAV-mediated gene delivery to express the NIR range γ-secretase reporter: C99 720–670 biosensor (*Houser et al., 2020*). Using NIR confocal microscopy, we uncovered that γ-secretase activity influences the activity of γ-secretase in neighboring neurons, suggesting a potential 'cell non-autonomous' regulation of γ-secretase in mouse brains.

Whether the γ-secretase activity is similarly or differently regulated among neurons remains elusive. A previous study employing microengraving technology reported that individual neurons generate and secrete different levels of Aβ peptides (*Liao et al., 2016*). In the same line, our FRET-based biosensors have allowed 'visualizing' that γ-secretase activity is heterogeneously regulated on a cell-by-cell basis in primary neurons (*Maesako et al., 2020*). Furthermore, the cell-by-cell heterogeneity in γ-secretase activity was further verified using (1) unique multiplexing FRET analysis in which the processing of two different substrates (e.g. APP vs. Notch1) by γ-secretase can be simultaneously measured in the same cell (*Houser et al., 2021*), and (2) multiplexed immunocytochemistry in which intracellular Aβ is detected on a cell-by-cell basis (*McKendell et al., 2022*). Our new study further adds that each neuron exhibits a distinct level of γ-secretase activity in intact mouse brains.

Although increasing evidence suggests γ-secretase activity is heterogeneously regulated on a cell-by-cell basis, little is known about the molecular mechanism(s) underlying such heterogeneity. Given that there are two isoforms of PSEN (i.e., PSEN1 and PSEN2) and three Aph1 (Aph1a, Aph1b, and Aph1c) in rodents, six different γ-secretase complexes can be expressed by the same cells. While the PSEN1 knockout mice displayed a lethal phenotype (*Shen et al., 1997*; *Wong et al., 1997*), the PSEN2 knockout revealed no severe phenotypes (*Herreman et al., 1999*). Furthermore, Aph1a knockout mice show a lethal phenotype, whereas those lacking Aph1b or Aph1c survive into adulthood (*Serneels et al., 2005*). These studies suggest that different γ-secretase complexes exhibit distinct γ-secretase activity. In this sense, the spatiotemporal expression of different γ-secretase complexes could explain the cell-by-cell basis heterogeneity in γ-secretase activity. On the other hand, previous studies utilizing pharmacological agents that bind to the active form of γ-secretase demonstrated that only a small portion of PSEN might be engaged in the active γ-secretase complex (*Lai et al., 2003*; *Placanica et al., 2009*). If this is the case, different equilibrium between the active and inactive γ-secretase could be another possible explanation of the heterogeneity in cellular γ-secretase activity.

Interestingly, we also uncovered not drastic but a statistically significant positive correlation between the 720/670 ratio and those ratios in neighboring neurons (*Figures 3 and 4*), suggesting that neurons with similar levels of γ-secretase activity form a cluster. Of note, this positive correlation in the γ-secretase activities is canceled by pharmacological inhibition of γ-secretase (*Figure 5*). Although whether DAPT has stochastic or differential accessibility to cells is a matter of further consideration, these results implicate that γ-secretase activity can 'propagate' from neuron to neuron. Nevertheless, the underlying mechanism remains elusive. Interestingly, a recent study shows that Aβ42 exerts product feedback inhibition on γ-secretase (*Zoltowska et al., 2024*), suggesting that secreted Aβ42 may be one of the negative regulators of γ-secretase. Furthermore, it is reported that hypoxia-inducible factor-1 alpha (HIF-1α), which is activated under hypoxia (*Semenza et al., 1991*; *Semenza and Wang,*

1992), regulates γ-secretase activity (*Villa et al., 2014*; *Alexander et al., 2022*). Therefore, local oxygen concentration may be linked to the clustering of neurons exhibiting similar levels of γ-secretase activity.

Conditional knockout of γ-secretase in the adult neurons displays synaptic dysfunctions, neuroinflammation, and neuronal loss in an age-dependent manner (*Saura et al., 2004*; *Tabuchi et al., 2009*; *Acx et al., 2017*; *Bi et al., 2021*). It is plausible that γ-secretase may function as 'a membrane proteasome' (*Kopan and Ilagan, 2004*), responsible for the degradation of over 150 different substrate stubs to maintain proper membrane homeostasis, which may be critical for neuronal survival. Notably, a conditional knockout strategy has also elucidated the critical roles of γ-secretase beyond the CNS, that is in other tissues, *in vivo*. For instance, specific PSEN1 knockout in hematopoietic progenitors demonstrated the importance of PSEN1/γ-secretase in developing and sustaining leukemia (*Habets et al., 2019*). Although GSIs were discontinued in the clinical trials of AD, their inhibiting effects on Notch signaling have led to the repurposing of GSIs as anticancer drugs (reviewed in *McCaw et al., 2021*). Furthermore, epidermis-specific Nicastrin conditional knockout allowed the identification of IL-36a as a key inflammatory cytokine involved in the malfunction of the skin barrier in the pathogenesis of HS (*Yang et al., 2020*). However, it is difficult to determine by existing tools how endogenous γ-secretase activity is spatiotemporally regulated and how the altered γ-secretase activity contributes to the disease progression. These dynamic processes in living cells can be directly monitored using our FRET-based imaging assays.

Lastly, in this study, we used NIR confocal microscopy to quantify γ-secretase activity in the superficial layers of the cortex (<200 micron in depth). Although multiphoton microscopy is the standard technique for *in vivo* imaging of the mouse brain, we believe our demonstration of NIR confocal microscopy of the living mouse brain also represents a unique and promising alternative technique, that avoids some of issues associated multiphoton microscopy including potential phototoxicity due to high average and peak laser powers and high complexity and costs of the instrumentation. For future studies aimed at interrogating γ-secretase activity in deeper cortical regions, multiphoton microscopy could be applied for Fluorescence lifetime imaging microscopy (FLIM) or ratiometric spectral imaging of either NIR (*Houser et al., 2020*) or visible FRET probes (*Maesako et al., 2020*).

In conclusion, this study provides a new imaging prototype to better understand the regulation of γ-secretase and its consequences in living mice *in vivo*. We have recorded γ-secretase activity on a cell-by-cell basis in mouse brains and found that neighboring neurons exhibit similar levels of γ-secretase activity, suggesting that γ-secretase influences the activity of γ-secretase in surrounding neurons.

## Materials and methods

### Adeno-associated virus (AAV)

Preparation of the AAV-hSyn1-C99 720–670 was performed as described previously (*Maesako et al., 2022*). Briefly, the cDNA of the C99 720–670 biosensor (*Houser et al., 2020*) was sub-cloned into a pAAV vector containing human Synapsin 1 promoter and WPRE sequences (*Maesako et al., 2017*). The plasmid sequence was verified by the MGH DNA core. The packaging into viruses (AAV2/8 stereotype) was performed at the University of Pennsylvania Gene Therapy Program vector core (Philadelphia, PA) (4.95E+13 GC/mL). The AAV-hSyn1-C99 720–670 was injected into the somatosensory cortex of 4–6 months old C57BL/6 male mice.

### Craniotomy

Craniotomy surgery was performed based on previously described methods (*Kuchibhotla et al., 2008*) with minor adjustments. Briefly, 4–6 months old male C57BL/6 mice (Charles River Laboratories, Wilmington, MA) were anesthetized using 1–1.5% isoflurane. A~3.5 mm circular section of the skull over the right hemisphere of the brain was surgically removed. The dura mater was left intact during the craniotomy procedure. A 4 mm cover glass was then positioned over the exposed brain area and secured with a mixture of dental acrylic and cyanoacrylate glue. Post-surgery, mice received buprenorphine and Tylenol for pain relief over 3 days. The mice were left to recover for at least 3 weeks before proceeding with the imaging experiments.

## NIR confocal microscopy

A diode laser at 640 nm wavelengths was used to excite the C99 720–670 biosensor. Fluorescence emission from miRFP670 (donor) and miRFP720 (acceptor) in the detection range: 670±10 nm and 750±50 nm, respectively, was detected by the high sensitivity-spectral detector, equipping cooled GaAsP photomultiplier on an Olympus FV3000RS confocal microscope. A x25 objective (NA = 1.05) was used for the image acquisition (512x512 pixels, Zoom x1). To obtain Z-stack images, the image acquisition was started at the first appearance of vasculature near the brain surface and continued at 2 µm step sizes up to approximately 60–80 sections.

## Image processing, quantification, and statistical analysis

For quantification, the 3D ImageJ Suite plugin (*Ollion et al., 2013*) in Fiji was used to automatically segment neurons expressing the C99 720–670 biosensor. In image processing, top-hat filtering was first applied to remove uneven background illumination (filter size: $r=7$) as described previously (*Netten et al., 1997*; *Gué et al., 2005*). Then, median filtering was used to remove noise from the images (filter size: $r=3$) (*Huang et al., 1979*). After the removal of the background and noise, 3D iterative thresholding (*Gul-Mohammed et al., 2014*) using the MSER criteria (*Matas et al., 2004*) was adapted to draw ROIs on the neuronal cell bodies (300–10000 voxels). Lastly, the morphological opening operation was applied to remove wrongly assigned ROIs (filter size: $r=2$; *Meyer and Beucher, 1990*). In order to determine γ-secretase activity on a cell-by-cell basis, the donor (miRFP670) and acceptor (miRFP720) fluorescence intensities in ROIs were measured, and the acceptor over donor emission ratios (i.e. 720/670 ratios) were calculated. Then, the distance between an ROI and other ROIs was measured using Scikit-learn (*Pedregosa et al., 2012*) in Python, which was used to identify neighboring neurons (e.g. 2, 5, 10 closest neurons, neurons within a 20 µm radius). Pseudo-color images corresponding to the 720/670 ratios were generated in MATLAB (MathWorks, Natick, MA).

In the statistical analysis, the Pearson correlation coefficient was measured to determine if the 720/670 ratios in a neuron significantly correlate with the 720/670 ratios of neighboring neurons, which was performed using Scipy (*Virtanen et al., 2020*) in Python. The codes written in MATLAB, and/or Python for imaging data processing and analysis will be shared upon requests. $p < 0.001$ was considered statistically significant. The sample size was calculated based on previous correlation analysis in our *in vitro* studies (*Houser et al., 2021*; *Maesako et al., 2022*) and was estimated to include approximately n=200–250 neurons/region-of-interests (ROIs) per animal. The number of biological replicates was shown in figures. All experiments were repeated in three independent experiments, and each experiment and analysis were replicated two times. The researchers who acquired and analyzed the data were blinded.

## Immunohistochemistry

Mice were euthanized using $CO_2$ asphyxiation and perfused with PBS, followed by 4% PFA (Electron Microscopy Sciences, Hatfield, PA). The extracted brains were postfixed by immersion in 4% PFA +15% glycerol (Sigma-Aldrich, St Louis, MO), and cryoprotected by 30% glycerol. Prior to immunostaining, the brains were sectioned using the Leica SM 2000R microtome (Bannockburn, IL) into 40 µm-thick coronal sections. Brain tissue sections were permeabilized using 0.4% Triton X-100 and blocked by incubation with 1.5% normal donkey serum (Jackson ImmunoResearch Labs, West Grove, PA). The free-floating sections were incubated with primary antibodies overnight at 4 °C. Anti-HA (RRID:AB_444303) and NeuN antibodies (RRID:AB_2532109) were purchased from Abcam (Cambridge, UK), anti-Iba-1 antibody (RRID:AB_839504) was from FUJIFILM Wako (Osaka, Japan), and anti-GFAP antibody (RRID:AB_477035) was from MilliporeSigma (Burlington, MA). The excess of the antibodies was washed off by PBS, and the sections were incubated with corresponding Alexa Fluor 488- or Cy3-conjugated secondary antibodies for 1 hr at room temperature. The brain sections were mounted with Fluoromount-G Mounting Medium, with DAPI (ThermoFisher, Waltham, MA).

## Acknowledgements

We thank Ms. Wadzanai H Ndambakuwa (MGH Neurology) for technical support. This work was funded by BrightFocus Foundation grant A2019056F (MM) and the National Institute of Health grants AG079838 (MM), AG072046 (SSH), AG015379 (OB), and AG044486 (OB).

## Additional information

### Funding

| Funder | Grant reference number | Author |
| --- | --- | --- |
| National Institute on Aging | AG079838 | Masato Maesako |
| BrightFocus Foundation | A2019056F | Masato Maesako |
| National Institute on Aging | AG072046 | Steven S Hou |
| National Institute on Aging | AG015379 | Oksana Berezovska |
| National Institute on Aging | AG044486 | Oksana Berezovska |

The funders had no role in study design, data collection and interpretation, or the decision to submit the work for publication.

### Author contributions

Steven S Hou, Conceptualization, Resources, Data curation, Formal analysis, Validation, Investigation, Methodology, Writing – review and editing; Yuya Ikegawa, Resources, Data curation, Software, Formal analysis, Validation, Visualization, Methodology, Writing - original draft; Yeseo Kwon, Investigation; Natalia Wieckiewicz, Formal analysis, Investigation, Visualization; Mei CQ Houser, Resources, Investigation; Brianna Lundin, Resources; Brian J Bacskai, Methodology, Writing – review and editing; Oksana Berezovska, Funding acquisition, Writing – review and editing; Masato Maesako, Conceptualization, Data curation, Supervision, Funding acquisition, Validation, Visualization, Methodology, Project administration, Writing – review and editing

### Author ORCIDs

Oksana Berezovska ⓘ https://orcid.org/0000-0003-4898-5788
Masato Maesako ⓘ https://orcid.org/0000-0002-1970-2462

### Ethics

All of the experimental procedures were in compliance with the NIH guidelines for the use of animals in experiments and were approved by the Massachusetts General Hospital Animal Care and Use Committee (2003N000243).

Reviewer #1 (Public Review): https://doi.org/10.7554/eLife.96848.3.sa1
Reviewer #2 (Public Review): https://doi.org/10.7554/eLife.96848.3.sa2
Reviewer #3 (Public Review): https://doi.org/10.7554/eLife.96848.3.sa3
Author response https://doi.org/10.7554/eLife.96848.3.sa4

## Additional files

### Supplementary files

• MDAR checklist

### Data availability

All data generated and/or analyzed during this study are included in the manuscript and supporting files.

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
