## [Editor Report · eLife assessment]

Hou and colleagues describe the the use of a previously characterized FRET sensor for use in determining γ-secretase activity in the brain of living mice. In an approach that targeted the sensor to neurons, they observe patterns of fluorescent sensor readout suggesting clustered regions of secretase activity. These results once validated would be **valuable** in the field of Alzheimer's Disease research, yet further validation of the approach is required, as the current evidence provided is **inadequate** to support the conclusions.

---

## [Referee Report · Reviewer #1 (Public Review)]

Summary:

In their paper, Hou and co-workers explored the use of a FRET sensor for endogenous g-sec activity in vivo in the mouse brain. They used AAV to deliver the sensor to the brain for neuron specific expression and applied NIR in cranial windows to assess FRET activity; optimizing as well an imaging and segmentation protocol. In brief they observe clustered g-sec activity in neighboring cells arguing for a cell non-autonomous regulation of endogenous g-sec activity in vivo.

Strengths:

Mone.

Weaknesses:

Overall the authors provide a very limited data set and in fact only a proof of concept that their sensor can be applied in vivo. This is not really a research paper, but a technical note. With respect to their observation of clustered activity, they now provide an overview image, next to zoomed details. However, from these images one cannot conclude 'by eye' any clustering event. This aligns with the very low r values. All neurons in the field show variable activity and a clustering is not really evident from these examples. Even within a cluster, there is variability. The authors now confirm that expression levels are indeed variable but are independent from the ratio measurements. Further, they controlled for specificity by including DAPT treatments, but opposite to their own in vitro data (in primary neurons) the ratios increased. The authors argue that both distance and orientation can either decrease or increase ratios and that the use of this biosensor should be explored model-by-model. This doesn't really confer high confidence and may hinder other groups in using this sensor reliably.

Secondly, there is still no physiological relevance for this observation. The experiments are performed in wild-type mice, but it would be more relevant to compare this with a fadPSEN1 KI or a PSEN1cKO model to investigate the contribution of a gain of toxic function or LOF to the claimed cell non-autonomous activations. The authors acknowledge this shortcoming but argue that this is for a follow-up study.

For instance, they only monitor activity in cell bodies, and miss all info on g-sec activity in neurites and synapses: what is the relevance of the cell body associated g-sec and can it be used as a proxy for neuronal g-sec activity? If cells 'communicate' g-sec activities, I would expect to see hot spots of activity at synapses between neurons.

Without some more validation and physiologically relevant studies, it remains a single observation and rather a technical note paper, instead of a true research paper.

---

## [Referee Report · Reviewer #2 (Public Review)]

Summary:

The manuscript by Hou et al is a short technical report which details the potential use of a recently developed FRET based biosensor for gamma-secretase activity (Houser et al 2020) for in vivo imaging in the mouse brain. Gamma-secretase plays a crucial role in Alzheimer's disease pathology and therefore developing methodologies for precise in vivo measurements would be highly valuable to better understand AD pathophysiology in animal models.

The current version of the sensor utilizes a pair of far-red fluorescent proteins fused to a substrate of the enzyme. Using live imaging, it was previously demonstrated it is possible to monitor gamma-secretase activity in cultured cells. Notably, this is a variant of a biosensor that was previously described using CFP-YFP variants FRET pair (Maesako et al, iScience. 2020). The main claim and hypothesis for the manuscript is that IR excitation and emission has considerable advantages in terms of depth of penetration, as well as reduction in autofluorescence. These properties would make this approach potentially suitable to monitor cellular level dynamics of Gama-secretase in vivo.

The authors use confocal microscopy and show it is possible to detect fluorescence from single cortical cells. The paper described in detail technical information regarding imaging and analysis. The data presented details analysis of FRET ratio (FR) measurements within populations of cells. The authors claim it is possible to obtain reliable measurements at the level of individual cells. They compare the FR values across cells and mice and find a spatial correlation among neighboring cells. This is compared with data obtained after inhibition of endogenous gamma-secretase activity, which abolishes this correlation.

Strengths:

The authors describe in detail their experimental design and analysis for in vivo imaging of the reporter. The idea of using a far-red FRET sensor for in vivo imaging is novel and potentially useful to circumvent many of the pitfalls associated with intensity-based FRET imaging in complex biological environments (such as autofluorescence and scattering).

Weaknesses:

There are several critical points regarding the validation of this approach:

(1) Regarding the variability and spatial correlation- the dynamic range of the sensor previously reported in vitro is in the range of 20-30% change (Houser et al 2020) whereas the range of FR detected in vivo is between cells is significantly larger in this MS. This raises considerable doubts for specific detection of cellular activity

(2) One direct way to test the dynamic range of the sensor in vivo, is to increase or decrease endogenous gamma-secretase activity and to ensure this experimental design allows to accurately monitor gamma-secretase activity. In the previous characterization of the reporter (Hauser et al 2020), DAPT application and inhibition of gamma-secretase activity results in increased FR (Figures 2 and 3 of Houser et al). This is in agreement with the design of the biosensor, since FR should be inversely correlated with enzymatic activity. Here, the authors repeated the experiment, and surprisingly found an opposite effect, in which DAPT significantly reduced FR.

The authors maintain that this result could be due to differences in cell-types, However, this experiment was previously performed in cultures cortical neurons and many different cell types, as noted by the authors in their rebuttal.

Instead, I would argue that these results further highlight the concerns of using FR in vivo, since based on their own data, there is no way to interpret this quantification. If DAPT reduces FR, does this mean we should now interpret the results of higher FR corresponds to higher g-sec activity? Given a number of papers from the authors claiming otherwise, I do not understand how one can interpret the results as indicating a cell-specific effect.

In conclusion, without any ground truth, it is impossible to assess and interpret what FR measurements of this sensor in vivo mean. Therefore, the use of this approach as a way to study g-sec activity in vivo seems premature.

---

## [Referee Report · Reviewer #3 (Public Review)]

This paper builds on the authors' original development of a near infrared (NIR) FRET sensor by reporting in vivo real-time measurements for gamma-secretase activity in the mouse cortex. The in vivo application of the sensor using state-of-the-art techniques is supported by a clear description and straightforward data, and the project represents significant progress because so few biosensors work in vivo. Notably, the NIR biosensor is detectable to ~ 100 µm depth in the cortex. A minor limitation is that this sensor has a relatively modest ΔF as reported in Houser et al, which is an additional challenge for its use in vivo. Thus, the data is fully dependent on post-capture processing and computational analyses. This can unintentionally introduce biases but is not an insurmountable issue with the proper controls that the authors have performed here.

The following opportunity for improving the system didn't initially present itself until the authors performed an important test of the FRET sensor in vivo following DAPT treatment. The authors get credit for diligently reporting the unexpected decrease in 720/670 FRET ratio. In turn this has led to a suggestion that this sensor would benefit from a control that is insensitive to gamma-secretase activity. FRET influences that are independent of gamma-secretase activity could be distinguished by this control.

From previous results in cultured neurons, the authors expected an increase in FRET following DAPT treatment in vivo. These expectations fit with the sensor's mode-of-action because a block of gamma-secretase activity should retain the fluorophores in proximity. When the authors observed decreased FRET, the conclusion was that the sensor performs differently in different cellular contexts. However, a major concern is that mechanistically it is unclear how this could occur with this type of sensor. The relative orientation of fluorophores indeed can contribute to FRET efficiency in tension-based sensors. However, the proteolysis expected with gamma-secretase activity would release tension and orientation constraints. Thus, the major contributing FRET factor is expected to be distance, not orientation. Alternative possibilities that could inadvertently affect readouts include an additional DAPT target in vivo sequestering the inhibitor, secondary pH effects on FRET, photo-bleaching, or an unidentified fluorophore quencher in vivo stimulated by DAPT. Ultimately this new FRET sensor would benefit from a control that is insensitive to gamma-secretase activity. FRET influences that are independent of gamma-secretase activity could be distinguished by this control.

---

## [Author Response]

The following is the authors’ response to the current reviews.

**Reviewer #1 (Public Review):**
Overall the authors provide a very limited data set and in fact only a proof of concept that their sensor can be applied in vivo. This is not really a research paper, but a technical note. With respect to their observation of clustered activity, they now provide an overview image, next to zoomed details. However, from these images one cannot conclude 'by eye' any clustering event. This aligns with the very low r values. All neurons in the field show variable activity and a clustering is not really evident from these examples. Even within a cluster, there is variability. The authors now confirm that expression levels are indeed variable but are independent from the ratio measurements. Further, they controlled for specificity by including DAPT treatments, but opposite to their own in vitro data (in primary neurons) the ratios increased. The authors argue that both distance and orientation can either decrease or increase ratios and that the use of this biosensor should be explored model-by-model. This doesn't really confer high confidence and may hinder other groups in using this sensor reliably.Secondly, there is still no physiological relevance for this observation. The experiments are performed in wild-type mice, but it would be more relevant to compare this with a fadPSEN1 KI or a PSEN1cKO model to investigate the contribution of a gain of toxic function or LOF to the claimed cell non-autonomous activations. The authors acknowledge this shortcoming but argue that this is for a follow-up study.For instance, they only monitor activity in cell bodies, and miss all info on g-sec activity in neurites and synapses: what is the relevance of the cell body associated g-sec and can it be used as a proxy for neuronal g-sec activity? If cells 'communicate' g-sec activities, I would expect to see hot spots of activity at synapses between neurons.Without some more validation and physiologically relevant studies, it remains a single observation and rather a technical note paper, instead of a true research paper.

The effect size was small, as stated in the original and revised manuscripts and the point-by-point responses to the 1st round review. Such subtle effects will likely be challenging to detect by eye. However, our unbiased quantification allowed us to detect a statistically significant linear correlation between the 720/670 ratio in each neuron and the average ratio in neighboring neurons, which we have verified using many different approaches (Figure 3, Figure 3—figure supplement 2, and Figure 4), and the correlation was canceled by the administration of γ-secretase inhibitor (Figure 5). Such objective analysis made us more confident to conclude that γ-secretase affects g-secretase in neighboring neurons.

We would also like to make clear the design of the C99 720-670 biosensor. Both C99, the sensing domain that is cleaved by γ-secretase, and the anchoring domain fused to miRFP670 are integrated into the membrane (Figure 1A). Therefore, how these two domains with four transmembrane regions are embedded in the membrane should affect the orientation between the donor, miRFP670, and the acceptor, miRFP720. As noted in our point-by-point responses to the initial review, we have previously validated that pharmacological inhibition of γ-secretase significantly increases the FRET ratio in various cell lines, including CHO, MEF, BV2, SHSY5Y cells, and mouse cortical primary neurons (Maesako et al., 2020; Houser et al., 2020, and unpublished observations). On the other hand, FRET reduction by γ-secretase inhibition was found in mouse primary neurons derived from the cerebellum (unpublished observations) as well as the somatosensory cortex neurons *in vivo* (this study). While we could not use the exact same imaging set-up between cortical primary neurons *in vitro* and those *in vivo* due to different expression levels of the biosensor, we could do it for *in vitro* cortical primary neurons vs. *in vitro* cerebellum neurons. We found by the direct comparison that 720/670 ratios are significantly higher in the cerebellum than the cortex neurons even in the presence of 1 μM DAPT (Author response image 1), a concentration that nearly completely inhibits γ-secretase activity. This suggests a different integration and stabilization pattern of the sensing and anchoring domains in the C99 720-670 biosensor between the cortex and cerebellum primary neurons, and thus, orientation between the donor and acceptor varies in the two neuronal types. We expect a similar scenario between cortical primary neurons *in vitro* and those *in vivo.* Of note, we have recently demonstrated that the cortex and cerebellum primary neurons exhibit distinct membrane properties (Lundin and Wieckiewicz et al., 2024 *in revision*), suggesting the different baseline FRET could be related to the different membrane properties between the cortex and cerebellum primary neurons. On the other hand, this raises a concern that 720/670 ratios can be affected not only by γ-secretase activity but also by other cofounders, such as altered membrane properties. However, a small but significant correlation between the 720/670 ratio in a neuron and those ratios in its neighboring neurons is canceled by γ-secretase inhibitor (Figure 5), suggesting that the correlation between the 720/670 ratio in a neuron and those in its neighboring neurons is most likely dependent on γ-secretase activity. Taken together, we *currently* think orientation plays a significant role in our biosensor and would like to emphasize the importance of ensuring on a model-by-model basis whether the cleavage of the C99 720-670 biosensor by γ-secretase increases or decreases 720/670 FRET ratios.

**Author response image 1. sa4fig1:** 

Furthermore, we co-expressed the C99 720-670 biosensor and visible range fluorescence reporters to record other biological events, such as changes in ion concentration, in cortex primary neurons. Interestingly, several biological events uniquely detected in the neurons with higher 720/670 ratios, which are expected to exhibit lower endogenous γ-secretase activity, are recapitulated by pharmacological inhibition of γ-secretase (unpublished observations), ensuring that higher 720/670 ratios are indicative of lower γ-secretase activity in mouse cortex primary neurons. Such multiplexed imaging will help to further elucidate how the C99 720-670 biosensor behaves in response to the modulation of γ-secretase activity.

Lastly, the scope of this study was to develop and validate a novel imaging assay employing a NIR FRET biosensor to measure γ-secretase activity on a cell-by-cell basis in live wild-type mouse brains. However, we do appreciate the reviewer’s suggestion and think employing this new platform in FAD PSEN1 knock-in (KI) or PSEN1 conditional knockout (cKO) mice would provide valuable information. Furthermore, we are keen to expand our capability to monitor γ-secretase with subcellular resolution in live mouse brains *in vivo,* which we will explore in follow-up studies. Thank you for your thoughtful suggestions.

Reference

- Maesako M, Sekula NM, Aristarkhova A, Feschenko P, Anderson LC, Berezovska O. Visualization of PS/γ-Secretase Activity in Living Cells. iScience. 2020 Jun 26;23(6):101139.

- Houser MC, Hou SS, Perrin F, Turchyna Y, Bacskai BJ, Berezovska O, Maesako M. A Novel NIR-FRET Biosensor for Reporting PS/γ-Secretase Activity in Live Cells. Sensors (Basel). 2020 Oct 22;20(21):5980.

- Lundin B, Wieckiewicz N, Dickson JR, Sobolewski RGR, Sadek M, Armagan G, Perrin F, Hyman BT, Berezovska O, and Maesako M. APP is a regulator of endo-lysosomal membrane permeability. 2024 *in revision*

**Reviewer #2 (Public Review):**
Regarding the variability and spatial correlation- the dynamic range of the sensor previously reported in vitro is in the range of 20-30% change (Houser et al 2020) whereas the range of FR detected in vivo is between cells is significantly larger in this MS. This raises considerable doubts for specific detection of cellular activity.One direct way to test the dynamic range of the sensor in vivo, is to increase or decrease endogenous gamma-secretase activity and to ensure this experimental design allows to accurately monitor gamma-secretase activity. In the previous characterization of the reporter (Hauser et al 2020), DAPT application and inhibition of gamma-secretase activity results in increased FR (Figures 2 and 3 of Houser et al). This is in agreement with the design of the biosensor, since FR should be inversely correlated with enzymatic activity. Here, the authors repeated the experiment, and surprisingly found an opposite effect, in which DAPT significantly reduced FR.The authors maintain that this result could be due to differences in cell-types, However, this experiment was previously performed in cultures cortical neurons and many different cell types, as noted by the authors in their rebuttal.Instead, I would argue that these results further highlight the concerns of using FR in vivo, since based on their own data, there is no way to interpret this quantification. If DAPT reduces FR, does this mean we should now interpret the results of higher FR corresponds to higher g-sec activity? Given a number of papers from the authors claiming otherwise, I do not understand how one can interpret the results as indicating a cell-specific effect.In conclusion, without any ground truth, it is impossible to assess and interpret what FR measurements of this sensor in vivo mean. Therefore, the use of this approach as a way to study g-sec activity in vivo seems premature.

Please find our response to reviewer 1’s similar critique above. Here, we again would like to re-clarify the design of our C99 720-670 biosensor. The orientation between the donor, miRFP670, and acceptor, miRFP720, is dependent on how C99, the sensing domain that is cleaved by γ-secretase, and the anchoring domain are integrated into the membrane (Figure 1A). Although it was surprising to us, it is possible that γ-secretase inhibition decreases 720/670 ratios if (1) the donor-acceptor orientation plays a significant role in FRET and (2) the baseline structure of the C99 720-670 biosensor is different between cell types. This appears to be the case between the cortex and cerebellum primary neurons (i.e., DAPT increases 720/670 ratios in the cortex neurons while decreasing in the cerebellum neurons), and we expect it in cortical neurons *in vitro* vs. *in vivo* as well. Hence, we recommend that users first validate whether the cleavage of the C99 720-670 biosensor by γ-secretase increases or decreases 720/670 FRET ratios in their models. If DAPT increases 720/670 ratios (like in cortex primary neurons, CHO, MEF, BV2, and SHSY5Y cells that we have validated), the results of higher ratios should be interpreted as lower γ-secretase activity. If DAPT reduces 720/670 ratios (like in cerebellum primary neurons and the somatosensory cortex neurons *in vivo*), we should interpret the results of higher ratios corresponding to higher γ-secretase activity. From a biosensing perspective, although we need to know which is the case on a model-by-model basis, we think whether γ-secretase activity increases or decreases the 720/670 ratio is not critical; rather, if it can significantly change FRET efficiency is more important. Thank you for your critical comments.

**Reviewer #3 (Public Review):**
This paper builds on the authors' original development of a near infrared (NIR) FRET sensor by reporting in vivo real-time measurements for gamma-secretase activity in the mouse cortex. The in vivo application of the sensor using state-of-the-art techniques is supported by a clear description and straightforward data, and the project represents significant progress because so few biosensors work in vivo. Notably, the NIR biosensor is detectable to ~ 100 µm depth in the cortex. A minor limitation is that this sensor has a relatively modest ΔF as reported in Houser et al, which is an additional challenge for its use in vivo. Thus, the data is fully dependent on post-capture processing and computational analyses. This can unintentionally introduce biases but is not an insurmountable issue with the proper controls that the authors have performed here.The following opportunity for improving the system didn't initially present itself until the authors performed an important test of the FRET sensor in vivo following DAPT treatment. The authors get credit for diligently reporting the unexpected decrease in 720/670 FRET ratio. In turn this has led to a suggestion that this sensor would benefit from a control that is insensitive to gamma-secretase activity. FRET influences that are independent of gamma-secretase activity could be distinguished by this control.From previous results in cultured neurons, the authors expected an increase in FRET following DAPT treatment in vivo. These expectations fit with the sensor's mode-of-action because a block of gamma-secretase activity should retain the fluorophores in proximity. When the authors observed decreased FRET, the conclusion was that the sensor performs differently in different cellular contexts. However, a major concern is that mechanistically it is unclear how this could occur with this type of sensor. The relative orientation of fluorophores indeed can contribute to FRET efficiency in tension-based sensors. However, the proteolysis expected with gamma-secretase activity would release tension and orientation constraints. Thus, the major contributing FRET factor is expected to be distance, not orientation. Alternative possibilities that could inadvertently affect readouts include an additional DAPT target in vivo sequestering the inhibitor, secondary pH effects on FRET, photo-bleaching, or an unidentified fluorophore quencher in vivo stimulated by DAPT. Ultimately this new FRET sensor would benefit from a control that is insensitive to gamma-secretase activity. FRET influences that are independent of gamma-secretase activity could be distinguished by this control.

Given that the anchoring domain is composed of three transmembrane regions and the linker connecting the donor, miRFP670, and the acceptor, miRFP720, is highly flexibility, we are still not sure if the orientation constraint of the C99 720-670 biosensor is canceled by γ-secretase cleavage. This means that the orientation between the donor and acceptor in the cleaved form of the sensor can be different between model and model. As explained in response to the similar critique of reviewer 1, we found that the 720/670 ratio is significantly higher in the cerebellum than in the cortex neurons even in the presence of DAPT (Figure 1 for the review only). Therefore, we *currently* think the donor-acceptor orientation, both in the cleaved and non-cleaved forms of the sensor, plays a role in determining whether γ-secretase activity increases or decreases the 720/670 ratio (but this view may change depends on the future discoveries).

As the reviewer pointed out, the NIR γ-secretase biosensor with no biological activity is important; however, a point mutation in the transmembrane region of the C99 sensing domain could also result in altered orientation between the donor, miRFP670, and the acceptor, miRFP720, since C99 is connected to the acceptor, which may bring additional complexity. Also, as noted in our point-by-point responses to the initial review, the mutation(s) that can fully block C99 processing by γ-secretase has not been established. Therefore, we asked if a subtle but significant correlation we found between the 720/670 ratio in a neuron and those ratios in its neighboring neurons is canceled by γ-secretase inhibitor administration. Since the correlation was abolished (Figure 5), it suggests that the correlation between the 720/670 ratio in a neuron and those ratios in the neighboring neurons depends on γ-secretase activity.

It is not fully established how -secretase activity is spatiotemporally regulated; therefore, the development of more appropriate control biosensors and further validation of our findings with complementary approaches would be crucial in our follow-up studies. Thank you for your valuable comments.

The following is the authors’ response to the original reviews.

**Reviewer #1 (Public Review):**
(1) Overall the authors provide a very limited data set and in fact only a proof of concept that their sensor can be applied in vivo. This is not really a research paper, but a technical note. With respect to their observation of clustered activity, the images do not convince me as they show only limited areas of interest: from these examples (for instance fig 5) one sees that merely all neurons in the field show variable activity and a clustering is not really evident from these examples. Even within a cluster, there is variability. With r values between 0.23 to .36, the correlation is not that striking. The authors herein do not control for expression levels of the sensor: for instance, can they show that in all neurons in the field, the sensor is equally expressed, but FRET activity is correlated in sets of neurons? Or are the FRET activities that are measured only in positively transduced neurons, while neighboring neurons are not expressing the sensor? Without such validation, it is difficult to make this conclusion.

We appreciate the reviewer’s comment. We agree with the reviewer that this study is not testing a new hypothesis but rather developing and validating a novel tool. However, we do believe such a “technical note” is as important as a “research paper” since advancing technique(s) is the only way to break the barrier in our understanding of complex biological events. Therefore, this study aimed to develop and validate a novel imaging assay employing a recently engineered NIR FRET biosensor to measure γ-secretase activity (Houser et al., 2020) on a cell-by-cell basis in live mouse brains, enabling us for the first time to examine how γ-secretase activity is regulated in individual neurons *in vivo*, and uncover that γ-secretase activity may influence γ-secretase in neighboring neurons. Like the reviewer, we found that the cell-to-cell correlation is not that striking, as we clearly stated in the original manuscript: “Although the effect size is modest, we also found a statistically significant correlation between…”

We were also aware that there is variability in a cluster of neurons exhibiting similar γ-secretase activities. Per the reviewer’s request, the images have been expanded to the entire imaging field of view (new Figure 3A). Although the effect size is small, our unbiased quantification showed a statistically significant linear correlation between the 720/670 ratio in each neuron and the average ratio in five neighboring neurons (Figure 3, Figure 3—figure supplement 2, and Figure 4), and the correlation was canceled by the administration of γ-secretase inhibitor (Figure 5). These findings made it impossible to conclude that γ-secretase does not affect γ-secretase in neighboring neurons.

Regarding the expression levels and pattern of the sensor, an AAV-based gene delivery approach employed in this study results in the expression of the sensor not in all but in selected neurons. We have newly performed immunohistochemistry, showing that approximately 40% of NeuN-positive neurons express the C99 720-670 biosensor (new Figure 1—figure supplement 2A and 2B).

Reference

- Houser MC, Hou SS, Perrin F, Turchyna Y, Bacskai BJ, Berezovska O, Maesako M. A Novel NIRFRET Biosensor for Reporting PS/γ-Secretase Activity in Live Cells. Sensors (Basel). 2020 Oct 22;20(21):5980.

(2) Secondly, I am lacking some more physiological relevance for this observation. The experiments are performed in wild-type mice, but it would be more relevant to compare this with a fadPSEN1 KI or a PSEN1cKO model to investigate the contribution of a gain of toxic function or LOF to the claimed cell non-autonomous activations. Or what would be the outcome if the sensor was targeted to glial cells?

The AAV vector in this study encodes the human synapsin promoter and our new immunohistochemistry demonstrates that nearly 100% of the cells expressing the C99 720-670 sensor are NeuN positive, and we hardly detected the sensor expression in Iba-1 or GFAP-positive cells (new Figure 1— figure supplement 2A and 2C).

The mechanism underlying the cell non-autonomous regulation of γ-secretase remains unclear. As discussed in our manuscript, one of the potential hypotheses could be that secreted abeta42 plays a role (Zoltowska et al., 2023 eLife). Whereas this report focuses on the development and validation of a novel assay using wildtype mice, future follow-up studies employing FAD PSEN1 knock-in (KI) and PSEN1 conditional knockout (cKO) mice would allow us test the hypothesis above since abeta42 is known to increase in some FAD PSEN1 KI mice (Siman et al., 2000 J Neurosci, Vidal et al., 2012 FASEB J) while decreases in PSEN1 cKO mice (Yu et al., 2001 Neuron).

Reference

- Siman R, Reaume AG, Savage MJ, Trusko S, Lin YG, Scott RW, Flood DG. Presenilin-1 P264L knockin mutation: differential effects on abeta production, amyloid deposition, and neuronal vulnerability. J Neurosci. 2000 Dec 1;20(23):8717-26.

- Vidal R, Sammeta N, Garringer HJ, Sambamurti K, Miravalle L, Lamb BT, Ghetti B. The Psen1-L166Pknock-in mutation leads to amyloid deposition in human wild-type amyloid precursor protein YAC transgenic mice. FASEB J. 2012 Jul;26(7):2899-910.

- Yu H, Saura CA, Choi SY, Sun LD, Yang X, Handler M, Kawarabayashi T, Younkin L, Fedeles B, Wilson MA, Younkin S, Kandel ER, Kirkwood A, Shen J. APP processing and synaptic plasticity in presenilin-1 conditional knockout mice. Neuron. 2001 Sep 13;31(5):713-26.

- Zoltowska KM, Das U, Lismont S, Enzlein T, Maesako M, Houser MC, Franco ML, Moreira DG, Karachentsev D, Becker A, Hopf C, Vilar M, Berezovska O, Mobley W, Chávez-Gutiérrez L. Alzheimer's disease linked Aβ42 exerts product feedback inhibition on γ-secretase impairing downstream cell signaling. eLife. 2023. 12:RP90690

(3) For this reviewer it is not clear what resolution they are measuring activity, at cellular or subcellular level? In other words are the intensity spots neuronal cell bodies? Given g-sec activity are in all endosomal compartments and at the cell surface, including in the synapse, does NIR imaging have the resolution to distinguish subcellular or surface localized activities? If cells 'communicate' g-sec activities, I would expect to see hot spots of activity at synapses between neurons: is this possible to assess with the current setup?

Since this study aimed to determine how γ-secretase activity is regulated on a cell-by-cell basis in live mouse brains, the FRET signal was detected in neuronal cell bodies. While our current set-up for *in vivo* can only record γ-secretase activity with a cellular resolution, we previously detected predominant γ-secretase activity in the endo-lysosomal compartments (Maesako et al., 2022 J Neurosci) as well as in certain spots of neuronal processes (Maesako et al., 2020 iScience) in *cultured* primary neurons using the same microscope set-up. Therefore, future studies will expand our capability to monitor γ-secretase with subcellular resolution in live mouse brains *in vivo.*

Reference

- Maesako M, Sekula NM, Aristarkhova A, Feschenko P, Anderson LC, Berezovska O. Visualization of PS/γ-Secretase Activity in Living Cells. iScience. 2020 Jun 26;23(6):101139.

- Maesako M, Houser MCQ, Turchyna Y, Wolfe MS, Berezovska O. Presenilin/γ-Secretase Activity Is Located in Acidic Compartments of Live Neurons. J Neurosci. 2022 Jan 5;42(1):145-154.

(4) Without some more validation and physiological relevant studies, it remains a single observation and rather a technical note paper, instead of a true research paper.

Please find our response above to the critique (1).

**Reviewer #2 (Public Review):**
(1) Regarding the variability and spatial correlation- the dynamic range of the sensor previously reported in vitro is in the range of 20-30% change (Houser et al 2020) whereas the range of FR detected in vivo is between cells is significantly larger (Fig. 3). This raises considerable doubts for specific detection of cellular activity (see point 3).

Please find our response below to the critique (2).

(2) One direct way to test the dynamic range of the sensor in vivo, is to increase or decrease endogenous gamma-secretase activity and to ensure this experimental design allows to accurately monitor gamma-secretase activity. In the previous characterization of the reporter (Hauser et al 2020), DAPT application and inhibition of gammasecretase activity results in increased FR (Figures 2 and 3 of Houser et al). This is in agreement with the design of the biosensor, since FR should be inversely correlated with enzymatic activity. Here, while the authors repeat the same manipulation and apply DAPT to block gamma-secretase activity, it seems to induce the opposite effect and reduces FR (comparing figures 8 with figures 5,6,7). First, there is no quantification comparing FR with and without DAPT. Moreover, it is possible to conduct this experiment in the same animals, meaning comparing FR before and after DAPT in the same mouse and cell populations. This point is absolutely critical- if indeed FR is reduced following DAPT application, this needs to be explained since this contradicts the basic design and interpretation of the biosensor.

We appreciate the reviewer’s comment. In our hand, overexpression of γ-secretase four components (PSEN, Nct, Aph1, and Pen2) is the only reliable and reproducible approach to increase the cellular activity of γ-secretase, which we successfully employed *in vitro* but not *in vivo* yet. Therefore, a γ-secretase inhibitor was used to determine the dynamic range of our FRET biosensor *in vivo*. FRET efficiency depends on the proximity and orientation of donor and acceptor fluorescent proteins. In our initial study, we engineered the original C99 EGFP-RFP biosensor (C99 R-G), and the replacement of EGFP and RFP with mTurquoise-GL and YPet, respectively, expanded the dynamic range of the sensor approximately 2 times. Moreover, extending the linker length from 20 a.a. to 80 a.a. increased the dynamic range 2.2 times (Maesako et al., 2020 iScience). Of note, the C99 720-670 NIR analog, which has the same 80 a.a. linker but miRFP670 and miRFP720 as the donor and acceptor, exhibited a slightly better dynamic range than the C99 Y-T sensor (Houser et al., 2020 Sensor). Our interpretation, at that time, was that the cleavage of the C99 720-670 biosensor by γ-secretase results in a longer distance between the donor and acceptor, and thus, the FRET ratio always increases by γ-secretase inhibition (i.e., proximity plays a more significant role than orientation in our biosensors). As expected, a significantly increased FRET ratio was detected in various cell lines by γ-secretase inhibitors, including CHO, MEF, BV2 cells, and mouse cortical primary neurons. Moreover, to further ensure the C99 720-670 biosensor records changes in γ-secretase activity, the multiplexing capability of the biosensor was utilized. In other words, we co-expressed the C99 720-670 biosensor and visible range fluorescence reporters to record other biological events, such as changes in ion concentration, etc., in cortex primary neurons. Strikingly, several biological events uniquely detected in the neurons with diminished endogenous γ-secretase activity, i.e., neurons with higher FRET ratios, are recapitulated by pharmacological inhibition of γ-secretase (unpublished observation). This approach has allowed us to ensure that increased FRET ratios are indicative of decreased endogenous γ-secretase activity in mouse cortical primary neurons.

However, as recommended by the reviewer, we have performed a new experiment to compare the FRET ratio before and after DAPT, a potent γ-secretase inhibitor, administration in the same mouse and cell populations. Surprisingly, we found that of DAPT significantly *decreases* 720/670 ratios, which is included in our revised manuscript (Figure 2—figure supplement 2C). This unexpected FRET reduction by γ-secretase inhibition was also found in mouse primary neurons derived from the cerebellum (unpublished observation). These findings suggest that orientation plays a significant role in our γ-secretase FRET biosensor and whether the FRET ratio is increased or decreased by the γ-secretase-mediated cleavage depends on cell types. Of note, the difference in FRET ratios with and without DAPT was comparable between primary cortex neurons (24.3%) and the somatosensory cortex neurons *in vivo* (22.1%). Our new findings suggest that how our biosensors report γ-secretase activity (i.e., increased vs. decreased FRET ratio) must be examined on a model-by-model basis, which is clearly noted in the revised manuscript:

Reference

- Houser MC, Hou SS, Perrin F, Turchyna Y, Bacskai BJ, Berezovska O, Maesako M. A Novel NIRFRET Biosensor for Reporting PS/γ-Secretase Activity in Live Cells. Sensors (Basel). 2020 Oct 22;20(21):5980.

- Maesako M, Sekula NM, Aristarkhova A, Feschenko P, Anderson LC, Berezovska O. Visualization of PS/γ-Secretase Activity in Living Cells. iScience. 2020 Jun 26;23(6):101139.

(3) For further validation, I would suggest including in vivo measurements with a sensor version with no biological activity as a negative control, for example, a mutation that prevents enzymatic cleavage and FRET changes. This should be used to showcase instrumental variability and would help to validate the variability of FR is indeed biological in origin. This would significantly strengthen the claims regarding spatial correlation within population of cells.

We fully agree with the reviewer that having a sensor version containing a mutation, which prevents enzymatic cleavage and thus FRET changes, as a negative control is preferable. In our previous study, we developed and validated the APP-based C99 Y-T and Notch1-based N100 Y-T biosensors (Maesako et al., 2020 iScience). It is well established that Notch1 cleavage is entirely blocked by Notch1 V1744G mutation (Schroeter et al., 1998 Nature; Huppert et al., 2000 Nature), and therefore, we introduced the mutation into N100 Y-T biosensor and used it as a negative control. On the other hand, such a striking mutation has never been identified in APP processing. To successfully monitor γ-secretase activity in deep tissue *in vivo*, we replaced Turquoise-GL and YPet in the C99 Y-T and N100 Y-T biosensors with miRFP670 and miRFP720, respectively. While the APP-based C99 720-670 biosensor allows recording γ-secretase activity (Houser et al., 2020 Sensors), we found the N100 720-670 sensor exhibits a very small dynamic range, not enabling to reliably measure γ-secretase activity. Taken together, there is not currently available NIR γ-secretase biosensor with no biological activity.

Reference

- Houser MC, Hou SS, Perrin F, Turchyna Y, Bacskai BJ, Berezovska O, Maesako M. A Novel NIRFRET Biosensor for Reporting PS/γ-Secretase Activity in Live Cells. Sensors (Basel). 2020 Oct 22;20(21):5980.

- Huppert SS, Le A, Schroeter EH, Mumm JS, Saxena MT, Milner LA, Kopan R. Embryonic lethality in mice homozygous for a processing-deficient allele of Notch1. Nature. 2000 Jun 22;405(6789):966-70.

- Maesako M, Sekula NM, Aristarkhova A, Feschenko P, Anderson LC, Berezovska O. Visualization of PS/γ-Secretase Activity in Living Cells. iScience. 2020 Jun 26;23(6):101139.

- Schroeter EH, Kisslinger JA, Kopan R. Notch-1 signalling requires ligand-induced proteolytic release of intracellular domain. Nature. 1998 May 28;393(6683):382-6.

(4) In general, confocal microcopy is not ideal for in vivo imaging. Although the authors demonstrate data collected using IR imaging increases penetration depth, out of focus fluorescence is still evident (Figure 4). Many previous papers have primarily used FLIM based analysis in combination with 2p microscopy for in vivo FRET imaging (Some examples: Ma et al, Neuron, 2018; Massengil et al, Nature methods, 2022; DIaz-Garcia et al, Cell Metabolism, 2017; Laviv et al, Neuron, 2020). This technique does not rely on absolute photon number and therefore has several advantage sin terms of quantification of FRET signals in vivo.It is therefore likely that use of previously developed sensors of gamma-secretase with conventional FRET pairs, might be better suited for in vivo imaging. This point should be at least discussed as an alternative.

The reviewer notes that 2p-FLIM may provide certain advantages over our confocal spectral imaging approach for detecting *in vivo* FRET. In our response below, we will address both the FRET detection method (FLIM vs. spectral) and microscope modality (2p vs. confocal).

As noted by the reviewer, we do acknowledge that 2p-FLIM has been utilized to detect FRET *in vivo*. On the other hand, the ratiometric spectral FRET approach has also been utilized in many *in vivo* FRET studies (Kuchibhotla et al., 2008 Neuron; Kuchibhotla et al., 2014 PNAS; Hiratsuka et al., 2015 eLife; Maesako et al., 2017 eLife; Konagaya et al., 2017 Cell Rep; Calvo-Rodriguez et al., 2020 Nat Communi; Hino et al., 2022 Dev Cell). We think both approaches have advantages and disadvantages, as discussed in a previous review (Bajar et al., 2016 Sensors), but they complement each other. Indeed, we regularly employ FLIM in cell culture studies (Maesako et al., 2017 eLife; McKendell et al., 2022 Biosensors; Devkota 2024 Cell Rep), and our recent study also utilized 2p-FLIM for *in vivo* NIR imaging (although not for detecting FRET) (Hou et al., 2023, Nat Biomed Eng); therefore, we are confident that 2p-FLIM can be adapted in our follow-up studies for γ-secretase recording.

Regarding microscope modality, we agree with the reviewer’s point that generally two-photon microscopy can achieve larger penetration depths than confocal microscopy and is therefore more ideal for *in vivo* FRET imaging. However, in this study, since our aim was to quantify γ-secretase activity in the superficial layers of the cortex (<200 microns in depth), both NIR confocal and multiphoton microscopies could be used to achieve this imaging objective. Additionally, we chose to use confocal microscopy with our NIR C99 720-670 probe due to the probe’s slightly but higher sensitivity compared to our C99 Y-T probe (Houser et al., 2020 Sensors). Imaging γ-secretase activity with our NIR C99-720-670 probe has the additional advantage that it will allow us in future studies to multiplex with visible FRET pairs using multiphoton microscopy in the same brain region. Furthermore, our demonstration of *in vivo* FRET imaging using NIR confocal microscopy avoids some of the issues associated with multiphoton microscopy, including potential phototoxicity due to high average and peak laser powers and the high complexity and costs of the instrumentation. For future studies aimed at interrogating γ-secretase activity in deeper cortical regions, multiphoton microscopy could be applied for FLIM or ratiometric spectral imaging of either our NIR or visible FRET probes. Per the reviewer’s request, we have added multiphoton FRET imaging as an alternative in the discussion section.

Reference

- Bajar BT, Wang ES, Zhang S, Lin MZ, Chu J. A Guide to Fluorescent Protein FRET Pairs. Sensors (Basel). 2016 Sep 14;16(9):1488.

- Calvo-Rodriguez M, Hou SS, Snyder AC, Kharitonova EK, Russ AN, Das S, Fan Z, Muzikansky A,

Garcia-Alloza M, Serrano-Pozo A, Hudry E, Bacskai BJ. Increased mitochondrial calcium levels

associated with neuronal death in a mouse model of Alzheimer's disease. Nat Commun. 2020 May

1;11(1):2146

- Devkota S, Zhou R, Nagarajan V, Maesako M, Do H, Noorani A, Overmeyer C, Bhattarai S, Douglas JT, Saraf A, Miao Y, Ackley BD, Shi Y, Wolfe MS. Familial Alzheimer mutations stabilize synaptotoxic γ-secretase-substrate complexes. Cell Rep. 2024 Feb 27;43(2):113761.

- Hino N, Matsuda K, Jikko Y, Maryu G, Sakai K, Imamura R, Tsukiji S, Aoki K, Terai K, Hirashima T, Trepat X, Matsuda M. A feedback loop between lamellipodial extension and HGF-ERK signaling specifies leader cells during collective cell migration. Dev Cell. 2022 Oct 10;57(19):2290-2304.e7.

- Hiratsuka T, Fujita Y, Naoki H, Aoki K, Kamioka Y, Matsuda M. Intercellular propagation of extracellular signal-regulated kinase activation revealed by in vivo imaging of mouse skin. eLife. 2015 Feb 10;4:e05178.

- Hou SS, Yang J, Lee JH, Kwon Y, Calvo-Rodriguez M, Bao K, Ahn S, Kashiwagi S, Kumar ATN, Bacskai BJ, Choi HS. Near-infrared fluorescence lifetime imaging of amyloid-β aggregates and tau fibrils through the intact skull of mice. Nat Biomed Eng. 2023 Mar;7(3):270-280.

- Houser MC, Hou SS, Perrin F, Turchyna Y, Bacskai BJ, Berezovska O, Maesako M. A Novel NIRFRET Biosensor for Reporting PS/γ-Secretase Activity in Live Cells. Sensors (Basel). 2020 Oct 22;20(21):5980.

- Konagaya Y, Terai K, Hirao Y, Takakura K, Imajo M, Kamioka Y, Sasaoka N, Kakizuka A, Sumiyama K, Asano T, Matsuda M. A Highly Sensitive FRET Biosensor for AMPK Exhibits Heterogeneous AMPK Responses among Cells and Organs. Cell Rep. 2017 Nov 28;21(9):2628-2638.

- Kuchibhotla KV, Goldman ST, Lattarulo CR, Wu HY, Hyman BT, Bacskai BJ. Abeta plaques lead to aberrant regulation of calcium homeostasis in vivo resulting in structural and functional disruption of neuronal networks. Neuron. 2008 Jul 31;59(2):214-25

- Kuchibhotla KV, Wegmann S, Kopeikina KJ, Hawkes J, Rudinskiy N, Andermann ML, Spires-Jones TL, Bacskai BJ, Hyman BT. Neurofibrillary tangle-bearing neurons are functionally integrated in cortical circuits in vivo. Proc Natl Acad Sci U S A. 2014 Jan 7;111(1):510-4

- Maesako M, Horlacher J, Zoltowska KM, Kastanenka KV, Kara E, Svirsky S, Keller LJ, Li X, Hyman BT, Bacskai BJ, Berezovska O. Pathogenic PS1 phosphorylation at Ser367. Elife. 2017 Jan 30;6:e19720.

- McKendell AK, Houser MCQ, Mitchell SPC, Wolfe MS, Berezovska O, Maesako M. In-Depth

Characterization of Endo-Lysosomal Aβ in Intact Neurons. Biosensors (Basel). 2022 Aug 20;12(8):663.

**(Recommendations For The Authors):**
(5) Minor issues- Figure 4 describes the analysis procedure, which seems to be standard practice in the field. This can be described in the methods section rather than in the main figure.

Per the reviewer’s suggestion, this figure has been moved to Figure 2—figure supplement 1.

**Reviewer #3 (Public Review):**
(1) This paper builds on the authors' original development of a near infrared (NIR) FRET sensor by reporting in vivo real-time measurements for gamma-secretase activity in the mouse cortex. The in vivo application of the sensor using state of the art techniques is supported by a clear description and straightforward data, and the project represents significant progress because so few biosensors work in vivo. Notably, the NIR biosensor is detectable to ~ 100 µm depth in the cortex. A minor limitation is that this sensor has a relatively modest ΔF as reported in Houser et al, which is an additional challenge for its use in vivo. Thus, the data is fully dependent on post-capture processing and computational analyses. This can unintentionally introduce biases but is not an insurmountable issue with the proper controls that the authors have performed here.

We appreciate the reviewer’s overall positive evaluation. As described in our response to the Reviewer 2’s critique (2), ΔF *in vivo* has been characterized (Figure 2—figure supplement 2C).

(2) The observation of gamma-secretase signaling that spreads across cells is potentially quite interesting, but it can be better supported. An alternative interpretation is that there exist pre-formed and clustered hubs of high gamma-secretase activity, and that DAPT has stochastic or differential accessibility to cells within the cluster. This could be resolved by an experiment of induction, for example, if gamma-secretase activity is induced or activated at a specific locale and there was observed coordinated spreading to neighboring neurons with their sensor.

We agree with the reviewer that the stochastic or differential accessibility of DAPT to cell clusters with different γ-secretase can be an alternative interpretation of our data, which is now included in the Discussion of the revised manuscript. Undoubtedly, the activation of γ-secretase would provide valuable information. However, as described in the response above to Reviewer 2’s critique #2, overexpressing the four components of γ-secretase (PSEN, Nct, Aph1, and Pen2) is the only reliable and reproducible approach to increasing the cellular activity of γ-secretase, which was achieved in our *in vitro* study but not yet *in vivo*. Our future study will develop and characterize the approach to induce γ-secretase activity to further perform detailed mechanistic studies.

(3) Furthermore, to rule out the possibility that uneven viral transduction was not simply responsible for the observed clustering, it would be helpful to see an analysis of 670nm fluorescence alone.

Our new analysis comparing 670 nm fluorescence intensity and that in five neighbor neurons shows a positive correlation (Figure 3—figure supplement 1A), suggesting that AAV was unevenly transduced. On the other hand, the 720/670 ratio (i.e., γ-secretase activity) is not correlated with 670 nm fluorescence intensity (i.e., C99 720-670 biosensor expression) (Figure 3—figure supplement 1B). This strongly suggests that, while C99 720-670 biosensor expression was not evenly distributed in the brain, the uneven probe expression did not impact the capability of γ-secretase recording.

**Reviewer #3 (Recommendations For The Authors):**
(4) One minor suggestion might be to consider Figures 6-7 as orthogonal supporting analyses rather than "validation". It might then be helpful to present them together with Figure 5.

We have moved the initial Figure 6 and 7 to Figure 3—figure supplement 2 and Figure 4, respectively.